# Learning a Natural Language Interface with Neural Programmer

**Arvind Neelakantan**[*]
University of Massachusetts Amherst
arvind@cs.umass.edu

**Quoc V. Le**
Google Brain
qvl@google.com

**Martín Abadi**
Google Brain
abadi@google.com

**Andrew McCallum**[*]
University of Massachusetts Amherst
mccallum@cs.umass.edu

**Dario Amodei**[*]
OpenAI
damodei@openai.com

## Abstract

Learning a natural language interface for database tables is a challenging task that involves deep language understanding and multi-step reasoning. The task is often approached by mapping natural language queries to *logical forms* or *programs* that provide the desired response when executed on the database. To our knowledge, this paper presents the first weakly supervised, end-to-end neural network model to induce such programs on a real-world dataset. We enhance the objective function of Neural Programmer, a neural network with built-in discrete operations, and apply it on WikiTableQuestions, a natural language question-answering dataset. The model is trained end-to-end with weak supervision of question-answer pairs, and does not require domain-specific grammars, rules, or annotations that are key elements in previous approaches to program induction. The main experimental result in this paper is that a single Neural Programmer model achieves 34.2% accuracy using only 10,000 examples with weak supervision. An ensemble of 15 models, with a trivial combination technique, achieves 37.7% accuracy, which is competitive to the current state-of-the-art accuracy of 37.1% obtained by a traditional natural language semantic parser.

## 1 Background and Introduction

Databases are a pervasive way to store and access knowledge. However, it is not straightforward for users to interact with databases since it often requires programming skills and knowledge about database schemas. Overcoming this difficulty by allowing users to communicate with databases via natural language is an active research area. The common approach to this task is by semantic parsing, which is the process of mapping natural language to symbolic representations of meaning. In this context, semantic parsing yields logical forms or programs that provide the desired response when executed on the databases (Zelle & Mooney, 1996). Semantic parsing is a challenging problem that involves deep language understanding and reasoning with discrete operations such as counting and row selection (Liang, 2016).

The first learning methods for semantic parsing require expensive annotation of question-program pairs (Zelle & Mooney, 1996; Zettlemoyer & Collins, 2005). This annotation process is no longer necessary in the current state-of-the-art semantic parsers that are trained using only question-answer pairs (Liang et al., 2011; Kwiatkowski et al., 2013; Krishnamurthy & Kollar, 2013; Pasupat & Liang, 2015). However, the performance of these methods still heavily depends on domain-specific grammar or pruning strategies to ease program search. For example, in a recent work on building semantic parsers for various domains, the authors hand-engineer a separate grammar for each domain (Wang et al., 2015).

Recently, many neural network models have been developed for program induction (Andreas et al., 2016; Jia & Liang, 2016; Reed & Freitas, 2016; Zaremba et al., 2016; Yin et al., 2015), despite

---

[*]Work done at Google Brain.

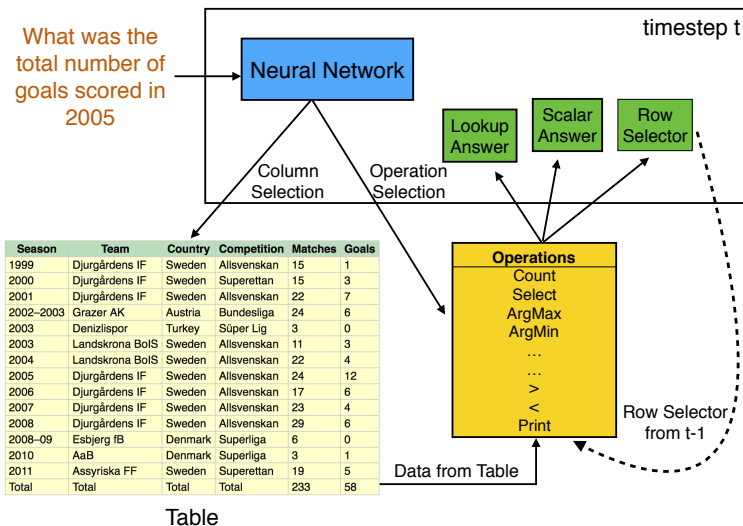

Figure 1: Neural Programmer is a neural network augmented with a set of discrete operations. The model runs for a fixed number of time steps, selecting an operation and a column from the table at every time step. The induced program transfers information across timesteps using the *row selector* variable while the output of the model is stored in the *scalar answer* and *lookup answer* variables.

the notorious difficulty of handling discrete operations in neural networks (Joulin & Mikolov, 2015; Kaiser & Sutskever, 2016). Most of these approaches rely on complete programs as supervision (Jia & Liang, 2016; Reed & Freitas, 2016) while others (Zaremba et al., 2016; Yin et al., 2015) have been tried only on synthetic tasks. The work that is most similar to ours is that of Andreas et al. (2016) on the *dynamic neural module network*. However, in their method, the neural network is employed only to search over a small set of candidate layouts provided by the syntactic parse of the question, and is trained using the REINFORCE algorithm (Williams, 1992). Hence, their method cannot recover from parser errors, and it is not trivial to adapt the parser to the task at hand. Additionally, all their modules or operations are parametrized by a neural network, so it is difficult to apply their method on tasks that require discrete arithmetic operations. Finally, their experiments concern a simpler dataset that requires fewer operations, and therefore a smaller search space, than WikiTableQuestions which we consider in our work. We discuss other related work in Section 4.

Neural Programmer (Neelakantan et al., 2016) is a neural network augmented with a set of discrete operations. It produces both a program, made up of those operations, and the result of running the program against a given table. The operations make use of three variables: *row selector*, *scalar answer*, and *lookup answer*, which are updated at every timestep. *lookup answer* and *scalar answer* store answers while *row selector* is used to propagate information across time steps. As input, a model receives a question along with a table (Figure 1). The model runs for a fixed number of time steps, selecting an operation and a column from the table as the argument to the operation at each time step. During training, soft selection (Bahdanau et al., 2014) is performed so that the model can be trained end-to-end using backpropagation. This approach allows Neural Programmer to explore the search space with better sample complexity than hard selection with the REINFORCE algorithm (Williams, 1992) would provide. All the parameters of the model are learned from a weak supervision signal that consists of only the final answer; the underlying program, which consists of a sequence of operations and of selected columns, is latent.

In this work, we develop an approach to semantic parsing based on Neural Programmer. We show how to learn a natural language interface for answering questions using database tables, thus integrating differentiable operations that are typical of neural networks with the declarative knowledge contained in the tables and with discrete operations on tables and entries. For this purpose, we make several improvements and adjustments to Neural Programmer, in particular adapting its objective function to make it more broadly applicable.

In earlier work, Neural Programmer is applied only on a synthetic dataset. In that dataset, when the expected answer is an entry in the given table, its position is explicitly marked in the table. However, real-world datasets certainly do not include those markers, and lead to many ambiguities (e.g., (Pasupat & Liang, 2015)). In particular, when the answer is a number that occurs literally in the table, it is not known, a priori, whether the answer should be generated by an operation or selected from the table. Similarly, when the answer is a natural language phrase that occurs in multiple positions in the table, it is not known which entry (or entries) in the table is actually responsible for the answer. We extend Neural Programmer to handle the weaker supervision signal by backpropagating through decisions that concern how the answer is generated when there is an ambiguity.

Our main experimental results concern WikiTableQuestions (Pasupat & Liang, 2015), a real-world question-answering dataset on database tables, with only 10,000 examples for weak supervision. This dataset is particularly challenging because of its small size and the lack of strong supervision, and also because the tables provided at test time are never seen during training, so learning requires adaptation at test time to unseen column names. A state-of-the-art, traditional semantic parser that relies on pruning strategies to ease program search achieves 37.1% accuracy. Standard neural network models like sequence-to-sequence and pointer networks do not appear to be promising for this dataset, as confirmed in our experiments below, which yield single-digit accuracies. In comparison, a single Neural Programmer model using minimal text pre-processing, and trained end-to-end, achieves 34.2% accuracy. This surprising result is enabled primarily by the sample efficiency of Neural Programmer, by the enhanced objective function, and by reducing overfitting via strong regularization with dropout (Srivastava et al., 2014; Iyyer et al., 2015; Gal & Ghahramani, 2016) and weight decay. An ensemble of 15 models, even with a trivial combination technique, achieves 37.7% accuracy.

## 2 NEURAL PROGRAMMER

In this section we describe in greater detail the Neural Programmer model and the modifications we made to the model. Neural Programmer is a neural network augmented with a set of discrete operations. The model consists of four modules:

- *Question RNN* that processes the question and converts the tokens to a distributed representation. We use an LSTM network (Hochreiter & Schmidhuber, 1997) as the question RNN.

- A list of discrete operations such as counting and entry selection that are manually defined. Each operation is parameterized by a real-valued vector that is learned during training.

- A *selector* module that induces two probability distributions at every time step, one over the set of operations and another over the set of columns. The input to the selector is obtained by concatenating the last hidden state of the question RNN, the hidden state of the history RNN from the current timestep, and the attention vector obtained by performing soft attention (Bahdanau et al., 2014) on the question using the history vector. Following Neelakantan et al. (2016), we employ hard selection at test time.

- *History RNN* modeled by a simple RNN (Werbos, 1990) with *tanh* activations which remembers the previous operations and columns selected by the model. The input to the history RNN at each timestep is the result of concatenating the weighted representations of operations and columns with their corresponding probability distributions produced by the selector at the previous timestep.

A more detailed description of the basic model can be found in Neelakantan et al. (2016). The model runs for fixed total of $T$ timesteps. The parameters of the operations, selector module, question and

history RNNs are all learned with backpropagation using a weak supervision signal that consists of the final answer. Below, we discuss several modifications to the model to make it more broadly applicable, and easier to train.

## 2.1 OPERATIONS

We use 15 operations in the model that were chosen to closely match the set of operations used in the baseline model (Pasupat & Liang, 2015). All the operations except *select* and *most frequent entry* operate only on the set of selected rows which is given by the *row selector* variable. Before the first timestep, all the rows in the table are set to be selected. The built-in operations are:

- *count* returns the number of selected rows in *row selector*.
- *select* and *most frequent entry* are operations which are computed only once for every question and output a boolean tensor with size same as the size of the input table. An entry in the output of the *select* operation is set to $1$ if the entry matches some phrase in the question. The matched phrases in the question are anonymized to prevent overfitting. Similarly, for *most frequent entry*, it is set to $1$ if the entry is the most frequently occurring one in its column.
- *argmax*, *argmin*, *greater than*, *less than*, *greater than or equal to*, *less than or equal to* are all operations that output a tensor with size same as the size of the input table.
- *first*, *last*, *previous* and *next* modify the *row selector*.
- *print* operation assigns *row selector* on the selected column of *lookup answer*.
- *reset* resets *row selector* to its initial value. This operation also serves as *no-op* when the model needs to induce programs whose complexity is less than $T$.

All the operations are defined to work with soft selection so that the model can be trained with backpropagation. The operations along with their definitions are discussed in the Appendix.

## 2.2 OUTPUT AND ROW SELECTOR

Neural programmer makes use of three variables: *row selector*, *scalar answer* and *lookup answer* which are updated at every timestep. The variable *lookup answer* stores answers that are selected from the table while *scalar answer* stores numeric answers that are not provided in the table.[1] The induced program transfers information across timesteps using the *row selector* variable which contains rows that are selected by the model.

Given an input table $\Pi$, containing $M$ rows and $C$ columns ($M$ and $C$ can vary across examples), the output variables at timestep $t$ are given by:

$scalar\ answer_t = \alpha_t^{op}(count)output_t(count),$

$lookup\ answer_t[i][j] = \alpha_t^{col}(j)\alpha_t^{op}(print)row\ select_{t-1}[i], \forall(i,j)i = 1,2,\ldots,M, j = 1,2,\ldots,C$

where $\alpha_t^{op}(op)$ and $\alpha_t^{col}(j)$ are the probabilities assigned by the selector to operation *op* and column $j$ at timestep $t$ respectively and $output_t(count)$ is the output of the count operation at timestep $t$. The row selector variable at timestep $t$ is obtained by taking the weighted average of the outputs of the remaining operations and is discussed in the Appendix. $lookup\ answer_T[i][j]$ is the probability that the element $(i,j)$ in the input table is in the final answer predicted by the model.

## 2.3 TRAINING OBJECTIVE

We modify the training objective of Neural Programmer to handle the supervision signal available in real-world settings. In previous work, the position of the answers are explicitly marked in the table when the answer is an entry from the table. However, as discussed in Section 1, in real-world datasets (e.g., (Pasupat & Liang, 2015)) the answer is simply written down introducing two kinds of ambiguities. First, when the answer is a number and if the number is in the table, it is not known

---

[1]It is possible to extend the model to generate natural language responses using an RNN decoder but it is not the focus of this paper and we leave it for further work.

whether the loss should be computed using the *scalar answer* variable or the *lookup answer* variable. Second, when the answer is a natural language phrase and if the phrase occurs in multiple positions in the table, we again do not know which entry (or entries) in the table is actually responsible for generating the answer. We extend Neural Programmer to handle this weaker supervision signal during training by computing the loss only on the prediction that is closest to the desired response.

For scalar answers we compute the square loss:

$$L_{scalar}(scalar\ answer_T, y) = \frac{1}{2}(scalar\ answer_T - y)^2$$

where $y$ is the ground truth answer. We divide $L_{scalar}$ by the number of rows in the input table and do not backpropagate on examples for which the loss is greater than a threshold since it leads to instabilities in training.

When the answer is a list of items $y = (a_1, a_2, \ldots, a_N)$, for each element in the list ($a_i, i = 1, 2, \ldots, N$) we compute all the entries in the table that match that element, given by $S_i = \{(r, c), \forall\ (r, c)\ \Pi[r][c] = a_i\}$. We tackle the ambiguity introduced when an answer item occurs at multiple entries in the table by computing the loss only on the entry which is assigned the highest probability by the model. We construct $g \in \{0, 1\}^{M \times C}$, where $g[i, j]$ indicates whether the element $(i, j)$ in the input table is part of the output. We compute log-loss for each entry and the final loss is given by:

$$L_{lookup}(lookup\ answer_T, y) = \sum_{i=1}^{N} min_{(r,c) \in S_i}(-\log(lookup\ answer_T[r, c]))$$

$$- \frac{1}{MC} \sum_{i=1}^{M} \sum_{j=1}^{C} [g[i, j] == 0] \log(1 - lookup\ answer_T[i, j])$$

where $[cond]$ is 1 when $cond$ is True, and 0 otherwise.

We deal with the ambiguity that occurs when the ground truth is a number and if the number also occurs in the table, by computing the final loss as the *soft minimum* of $L_{scalar}$ and $L_{lookup}$. Otherwise, the loss for an example is $L_{scalar}$ when the ground truth is a number and $L_{lookup}$ when the ground truth matches some entries in the table. The two loss functions $L_{scalar}$ and $L_{lookup}$ are in different scales, so we multiply $L_{lookup}$ by a constant factor which we set to 50.0 after a small exploration in our experiments.

Since we employ hard selection at test time, only one among *scalar answer* and *lookup answer* is modified at the last timestep. We use the variable that is set at the last timestep as the final output of the model.

## 3 EXPERIMENTS

We apply Neural Programmer on the WikiTableQuestions dataset (Pasupat & Liang, 2015) and compare it to different non-neural baselines including a natural language semantic parser developed by Pasupat & Liang (2015). Further, we also report results from training the sequence-to-sequence model (Sutskever et al., 2014) and a modified version of the pointer networks (Vinyals et al., 2015). Our model is implemented in TensorFlow (Abadi et al., 2016) and the model takes approximately a day to train on a single Tesla K80 GPU. We use double-precision format to store the model parameters since the gradients become undefined values in single-precision format. Our code is available at `https://github.com/tensorflow/models/tree/master/neural_programmer`.

### 3.1 DATA

We use the train, development, and test split given by Pasupat & Liang (2015). The dataset contains $11321$, $2831$, and $4344$ examples for training, development, and testing respectively. We use their tokenization, number and date pre-processing. There are examples with answers that are neither

| Method | Dev Accuracy | Test Accuracy |
|---|---|---|
| Baselines from Pasupat & Liang (2015) | | |
| Information Retrieval System | 13.4 | 12.7 |
| Simple Semantic Parser | 23.6 | 24.3 |
| Semantic Parser | 37.0 | 37.1 |
| Neural Programmer | | |
| Neural Programmer | 34.1 | 34.2 |
| Ensemble of 15 Neural Programmer models | 37.5 | 37.7 |
| Oracle Score with 15 Neural Programmer models | 50.5 | - |

Table 1: Performance of Neural Programmer compared to baselines from (Pasupat & Liang, 2015). The performance of an ensemble of 15 models is competitive to the current state-of-the-art natural language semantic parser.

number answers nor phrases selected from the table. We ignore these questions during training but the model is penalized during evaluation following Pasupat & Liang (2015). The tables provided in the test set are unseen at training, hence requiring the model to adapt to unseen column names at test time. We train only on examples for which the provided table has less than 100 rows since we run out of GPU memory otherwise, but consider all examples at test time.

## 3.2 TRAINING DETAILS

We use $T = 4$ timesteps in our experiments. Words and operations are represented as 256 dimensional vectors, and the hidden vectors of the question and the history RNN are also 256 dimensional. The parameters are initialized uniformly randomly within the range [-0.1, 0.1]. We train the model using the Adam optimizer (Kingma & Ba, 2014) with mini-batches of size 20. The $\epsilon$ hyperparameter in Adam is set to 1e-6 while others are set to the default values. Since the training set is small compared to other datasets in which neural network models are usually applied, we rely on strong regularization:

- We clip the gradients to norm 1 and employ early-stopping.
- The occurrences of words that appear less than 10 times in the training set are replaced by a single unknown word token.
- We add a weight decay penalty with strength 0.0001.
- We use dropout with a keep probability of 0.8 on input and output vectors of the RNN, and selector, operation and column name representations (Srivastava et al., 2014).
- We use dropout with keep probability of 0.9 on the recurrent connections of the question RNN and history RNN using the technique from Gal & Ghahramani (2016).
- We use word-dropout (Iyyer et al., 2015) with keep probability of 0.9. Here, words in the question are randomly replaced with the unknown word token while training.

We tune the dropout rates, regularization strength, and the $\epsilon$ hyperparameter using grid search on the development data, we fix the other hyperparameters after a small exploration during initial experiments.

## 3.3 RESULTS

Table 1 shows the performance of our model in comparison to baselines from Pasupat & Liang (2015). The best result from Neural Programmer is achieved by an ensemble of 15 models. The only difference among these models is that the parameters of each model is initialized with a different random seed. We combine the models by averaging the predicted softmax distributions of the models at every timestep. While it is generally believed that neural network models require a large number of training examples compared to simpler linear models to get good performance, our model

| Method | Dev Accuracy |
|---|---|
| Neural Programmer | 34.1 |
| Neural Programmer - anonymization | 33.7 |
| Neural Programmer - match feature | 31.1 |
| Neural Programmer - {dropout,weight decay} | 30.3 |

Table 2: Model ablation studies. We find that dropout and weight decay, along with the boolean feature indicating a matched table entry for column selection, have a significant effect on the performance of the model.

achieves competitive performance on this small dataset containing only 10,000 examples with weak supervision.

We did not get better results either by using pre-trained word vectors (Mikolov et al., 2013) or by pre-training the question RNN with a language modeling objective (Dai & Le, 2015). A possible explanation is that the word vectors obtained from unsupervised learning may not be suitable to the task under consideration. For example, the learned representations of words like *maximum* and *minimum* from unsupervised learning are usually close to each other but for our task it is counter-productive. We consider replacing soft selection with hard selection and training the model with the REINFORCE algorithm (Williams, 1992). The model fails to learn in this experiment, probably because the model has to search over millions of symbolic programs for every input question making it highly unlikely to find a program that gives a reward. Hence, the parameters of the model are not updated frequently enough.

### 3.3.1 Neural Network Baselines

To understand the difficulty of the task for neural network models, we experiment with two neural network baselines: the sequence-to-sequence model (Sutskever et al., 2014) and a modified version of the pointer networks (Vinyals et al., 2015). The input to the sequence-to-sequence model is a concatenation of the table and the question, and the decoder produces the output one token at a time. We consider only examples whose input length is less than 400 to make the running time reasonable. The resulting dataset has 8, 857 and 1, 623 training and development examples respectively. The accuracy of the best model on this development set after hyperparameter tuning is only 8.9%. Next, we experiment with pointer networks to select entries in the table as the final answer. We modify pointer networks to have two-attention heads: one to select the column and the other to select entries within a column. Additionally, the model performs multiple pondering steps on the table before returning the final answer. We train this model only on lookup questions, since the model does not have a decoder to generate answers. We consider only examples whose tables have less than 100 rows resulting in training and development set consisting of 7, 534 and 1, 829 examples respectively. The accuracy of the best model on this development set after hyperparameter tuning is only 4.0%. These results confirm our intuition that discrete operations are hard to learn for neural networks particularly with small datasets in real-world settings.

### 3.4 Analysis

### 3.4.1 Model Ablation

Table 2 shows the impact of different model design choices on the final performance. While anonymizing phrases in the question that match some table entry seems to have a small positive effect, regularization has a much larger effect on the performance. Column selection is performed in Neelakantan et al. (2016) using only the name of a column; however, this selection procedure is insufficient in real-world settings. For example the column selected in question 3 in Table 3 does not have a corresponding phrase in the question. Hence, to select a column we additionally use a boolean feature that indicates whether an entry in that column matches some phrase in the question. Table 2 shows that the addition of this boolean feature has a significant effect on performance.

| ID | Question | | Step 1 | Step 2 | Step 3 | Step 4 |
|---|---|---|---|---|---|---|
| 1 | what is the total number of teams? | Operation | - | - | - | count |
| | | Column | - | - | - | - |
| 2 | how many games had more than 1,500 in attendance? | Operation | - | - | >= | count |
| | | Column | - | - | attendance | - |
| 3 | what is the total number of runner-ups listed on the chart? | Operation | - | - | select | count |
| | | Column | - | - | outcome | - |
| 4 | which year held the most competitions? | Operation | - | - | mfe | print |
| | | Column | - | - | year | year |
| 5 | what opponent is listed last on the table? | Operation | last | - | last | print |
| | | Column | - | - | - | opponent |
| 6 | which section is longest?? | Operation | - | - | argmax | print |
| | | Column | - | - | kilometers | name |
| 7 | which engine(s) has the least amount of power? | Operation | - | - | argmin | print |
| | | Column | - | - | power | engine |
| 8 | what was claudia roll's time? | Operation | - | - | select | print |
| | | Column | - | - | swimmer | time |
| 9 | who had more silver medals, cuba or brazil? | Operation | argmax | select | argmax | print |
| | | Column | nation | nation | silver | nation |
| 10 | who was the next appointed director after lee p. brown? | Operation | select | next | last | print |
| | | Column | name | - | - | name |
| 11 | what team is listed previous to belgium? | Operation | select | previous | first | print |
| | | Column | team | - | - | team |

Table 3: A few examples of programs induced by Neural Programmer that generate the correct answer in the development set. mfe is abbreviation for the operation *most frequent entry*. The model runs for 4 timesteps selecting an operation and a column at every step. The model employs hard selection during evaluation. The column name is displayed in the table only when the operation picked at that step takes in a column as input while the operation is displayed only when it is other than the *reset* operation. Programs that choose *count* as the final operation produce a number as the final answer while programs that select *print* as the final operation produce entries selected from the table as the final answer.

| Operation | Program in Table 3 | Amount (%) |
|---|---|---|
| Scalar Answer | | |
| Only Count | 1 | 6.5 |
| Comparison + Count | 2 | 2.1 |
| Select + Count | 3 | 22.1 |
| Scalar Answer | 1,2,3 | 30.7 |
| Lookup Answer | | |
| Most Frequent Entry + Print | 4 | 1.7 |
| First/Last + Print | 5 | 9.5 |
| Superlative + Print | 6,7 | 13.5 |
| Select + Print | 8 | 17.5 |
| Select + {first, last, previous, next, superlative} + Print | 9-11 | 27.1 |
| Lookup Answer | 4-11 | 69.3 |

Table 4: Statistics of the different sequence of operations among the examples answered correctly by the model in the development set. For each sequence of operations in the table, we also point to corresponding example programs in Table 3. Superlative operations include *argmax* and *argmin*, while comparison operations include *greater than*, *less than*, *greater than or equal to* and *less than or equal to*. The model induces a program that results in a scalar answer 30.7% of the time while the induced program is a table lookup for the remaining questions. *print* and *select* are the two most common operations used 69.3% and 66.7% of the time respectively.

### 3.4.2 INDUCED PROGRAMS

Table 3 shows few examples of programs induced by Neural Programmer that yield the correct answer in the development set. The programs given in Table 3 show a few characteristics of the learned model. First, our analysis indicates that the model can adapt to unseen column names at test time. For example in Question 3, the word *outcome* occurs only 8 times in the training set and is replaced with the unknown word token. Second, the model does not always induce the most efficient (with respect to number of operations other than the *reset* operation that are picked) program to solve a task. The last 3 questions in the table can be solved using simpler programs. Finally, the model does not always induce the correct program to get the ground truth answer. For example, the last 2 programs will not result in the correct response for all input database tables. The programs would produce the correct response only when the *select* operation matches one entry in the table.

### 3.4.3 CONTRIBUTION OF DIFFERENT OPERATIONS

Table 4 shows the contribution of the different operations. The model induces a program that results in a scalar answer 30.7% of the time while the induced program is a table lookup for the remaining questions. The two most commonly used operations by the model are *print* and *select*.

### 3.4.4 ERROR ANALYSIS

To conclude this section, we suggest ideas to potentially improve the performance of the model. First, the oracle performance with 15 Neural Programmer models is 50.5% on the development set while averaging achieves only 37.5% implying that there is still room for improvement. Next, the accuracy of a single model on the training set is 53% which is about 20% higher than the accuracy in both the development set and the test set. This difference in performance indicates that the model suffers from significant overfitting even after employing strong regularization. It also suggests that the performance of the model could be greatly improved by obtaining more training data. Nevertheless, there are limits to the performance improvements we may reasonably expect: in particular, as shown in previous work (Pasupat & Liang, 2015), 21% of questions on a random set of 200 examples in the considered dataset are not answerable because of various issues such as annotation errors and tables requiring advanced normalization.

## 4 OTHER RELATED WORK

While we discuss in detail various semantic parsing and neural program induction techniques in Section 1, here we briefly describe other relevant work. Recently, Kocisky et al. (2016) develop a semi-supervised semantic parsing method that uses question-program pairs as supervision. Concurrently to our work, Liang et al. (2016) propose *neural symbolic machine*, a model very similar to Neural Programmer but trained using the REINFORCE algorithm (Williams, 1992). They use only 2 discrete operations and run for a total of 3 timesteps, hence inducing programs that are much simpler than ours. Neural networks have also been applied on question-answering datasets that do not require much arithmetic reasoning (Bordes et al., 2014; Iyyer et al., 2014; Sukhbaatar et al., 2015; Peng et al., 2015; Hermann et al., 2015; Kumar et al., 2016). Wang & Jiang (2016) use a neural network model to get state-of-the-art results on a reading comprehension task (Rajpurkar et al., 2016).

## 5 CONCLUSION

In this paper, we enhance Neural Programmer to work with weaker supervision signals to make it more broadly applicable. Soft selection during training enables the model to actively explore the space of programs by backpropagation with superior sample complexity. In our experiments, we show that the model achieves performance comparable to a state-of-the-art traditional semantic parser even though the training set contains only 10,000 examples. To our knowledge, this is the first instance of a weakly supervised, end-to-end neural network model that induces programs on a real-world dataset.

**Acknowledgements** We are grateful to Panupong Pasupat for answering numerous questions about the dataset, and providing pre-processed version of the dataset and the output of the semantic parser. We thank David Belanger, Samy Bengio, Greg Corrado, Andrew Dai, Jeff Dean, Nando de Freitas, Shixiang Gu, Navdeep Jaitly, Rafal Jozefowicz, Ashish Vaswani, Luke Vilnis, Yuan Yu and Barret Zoph for their suggestions and the Google Brain team for the support. Arvind Neelakantan is supported by a Google PhD fellowship in machine learning.

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

| Type | Operation | Definition |
|---|---|---|
| Aggregate | count | $count_t = \sum\limits_{i=1}^{M} row\_select_{t-1}[i]$ |
| Superlative | argmax | $max_t[i][j] = \max(0.0, row\_select_{t-1}[i] - \sum_{k=1}^{M}([\Pi[i][j] < \Pi[k][j]] \times row\_select_{t-1}[k])), i = 1, \ldots, M, j = 1, \ldots, C$ |
| | argmin | $min_t[i][j] = \max(0.0, row\_select_{t-1}[i] - \sum_{k=1}^{M}([\Pi[i][j] > \Pi[k][j]] \times row\_select_{t-1}[k])), i = 1, \ldots, M, j = 1, \ldots, C$ |
| Comparison | > | $g[i][j] = \Pi[i][j] > pivot_g, \forall(i,j), i = 1, \ldots, M, j = 1, \ldots, C$ |
| | < | $l[i][j] = \Pi[i][j] < pivot_l, \forall(i,j), i = 1, \ldots, M, j = 1, \ldots, C$ |
| | $\geq$ | $ge[i][j] = \Pi[i][j] \geq pivot_{ge}, \forall(i,j), i = 1, \ldots, M, j = 1, \ldots, C$ |
| | $\leq$ | $le[i][j] = \Pi[i][j] \leq pivot_{le}, \forall(i,j), i = 1, \ldots, M, j = 1, \ldots, C$ |
| Table Ops | select | $s[i][j] = 1.0$ if $\Pi[i][j]$ appears in question else $0.0$, $\forall(i,j), i = 1, \ldots, M, j = 1, \ldots, C$ |
| | mfe | $mfe[i][j] = 1.0$ if $\Pi[i][j]$ is the most common entry in column j else $0.0$, $\forall(i,j), i = 1, \ldots, M, j = 1, \ldots, C$ |
| | first | $f_t[i] = \max(0.0, row\_select_{t-1}[i] - \sum_{j=1}^{i-1} row\_select_{t-1}[j])$, $i = 1, \ldots, M$ |
| | last | $la_t[i] = \max(0.0, row\_select_{t-1}[i] - \sum_{j=i+1}^{M} row\_select_{t-1}[j])$, $i = 1, \ldots, M$ |
| | previous | $p_t[i] = row\_select_{t-1}[i+1], i = 1, \ldots, M-1 \, ; p_t[M] = 0$ |
| | next | $n_t[i] = row\_select_{t-1}[i-1], i = 2, \ldots, M \, ; n_t[1] = 0$ |
| Print | print | $lookup \, answer_t[i][j] = row\_select_{t-1}[i], \forall(i,j)i = 1, \ldots, M, j = 1, \ldots, C$ |
| Reset | reset | $r_t[i] = 1, \forall i = 1, 2, \ldots, M$ |

Table 5: List of all operations provided to the model along with their definitions. mfe is abbreviation for the operation *most frequent entry*. $[cond]$ is 1 when $cond$ is True, and 0 otherwise. Comparison, select, reset and mfe operations are independent of the timestep while all the other operations are computed at every time step. Superlative operations and most frequent entry are computed within a column. The operations calculate the expected output with the respect to the membership probabilities given by the row selector so that they can work with probabilistic selection.

## APPENDIX

### OPERATIONS

Table 5 shows the list of operations built into the model along with their definitions.

### ROW SELECTOR

As discussed in Section 2.3, the output variables *scalar answer* and *lookup answer* are calculated using the output of the count operations and print operation respectively. The *row selector* is computed using the output of the remaining operations and is given by,

$$
\begin{aligned}
row \, selector_t[i] = \sum_{j=1}^{C} & \{\alpha_t^{col}(j)\alpha_t^{op}(>)g[i][j] + \alpha_t^{col}(j)\alpha_t^{op}(<)l[i][j] \\
& + \alpha_t^{col}(j)\alpha_t^{op}(\geq)ge[i][j] + \alpha_t^{col}(j)\alpha_t^{op}(\leq)le[i][j], \\
& + \alpha_t^{col}(j)\alpha_t^{op}(argmax)max_t[i][j] + \alpha_t^{col}(j)\alpha_t^{op}(argmin_t)min[i][j], \\
& + \alpha_t^{col}(j)\alpha_t^{op}(select)s[i][j] + \alpha_t^{col}(j)\alpha_t^{op}(mfe)mfe[i][j]\} \\
& + \alpha_t^{op}(previous)p_t[i] + \alpha_t^{op}(next)n_t[i] + \alpha_t^{op}(reset)r_t[i] \\
& + \alpha_t^{op}(first)f_t[i] + \alpha_t^{op}(last)la_t[i] \\
& \forall i, i = 1, 2, \ldots, M
\end{aligned}
$$

where $\alpha_t^{op}(op)$ and $\alpha_t^{col}(j)$ are the probabilities assigned by the selector to operation *op* and column $j$ at timestep $t$ respectively.

