# Peer review of "Learning a Natural Language Interface with Neural Programmer"

_ICLR 2017 — accepted_

[Official Review · AnonReviewer5 · rating 6 · confidence 4 · 16 Dec 2016]
**A paper on a challenging task**

This paper proposes a weakly supervised, end-to-end neural network model to learn a natural language interface for tables. The neural programmer is applied to the WikiTableQuestions, a natural language QA dataset and achieves reasonable accuracy. An ensemble further boosts the performance by combining components built with different configurations, and achieves comparable performance as the traditional natural language semantic parser baseline. Dropout and weight decay seem to play a significant role.

It'll be interesting to see more error analysis and the major reason for the still low accuracy compared to many other NLP tasks. What's the headroom and oracle number with the current approach?

[Official Review · AnonReviewer3 · rating 6 · confidence 3 · 17 Dec 2016]

This paper proposes a weakly supervised, end-to-end neural network model for solving a challenging natural language understanding task. 
As an extension of the Neural Programmer, this work aims at overcoming the ambiguities imposed by natural language. 
By predefining a set of operations, the model is able to learn the interface between the language reasoning and answer composition using backpropagation. 
On the WikiTableQuestions dataset, it is able to achieve a slightly better performance than the traditional semantic parser methods. 

Overall, this is a very interesting and promising work as it involves a lot of real-world challenges about natural language understanding. 
The intuitions and design of the model are very clear, but the complication makes the paper a bit difficult to read, which means the model is also difficult to be reimplemented. I would expect to see more details about model ablation and it would help us figure out the prominent parts of the model design.

[Official Review · AnonReviewer4 · rating 7 · confidence 3 · 19 Dec 2016]
**An interesting paper for a rather hard problem.**

The paper presents an end-to-end neural network model for the problem of designing natural language interfaces for database queries. The proposed approach uses only weak supervision signals to learn the parameters of the model. Unlike in traditional approaches, where the problem is solved by semantically parsing a natural language query into logical forms and executing those logical forms over the given data base, the proposed approach trains a neural network in an end-to-end manner which goes directly from the natural language query to the final answer obtained by processing the data base. This is achieved by formulating a collection of operations to be performed over the data base as continuous operations, the distributions over which is learnt using the now-standard soft attention mechanisms. The model is validated on the smallish WikiTableQuestions dataset, where the authors show that a single model performs worse than the approach which uses the traditional Semantic Parsing technique. However an ensemble of 15 models (trained in a variety of ways) results in comparable performance to the state of the art. 

I feel that the paper proposes an interesting solution to the hard problem of learning natural language interfaces for data bases. The model is an extension of the previously proposed models of Neelakantan 2016. The experimental section is rather weak though. The authors only show their model work on a single smallish dataset. Would love to see more ablation studies of their model and comparison against fancier version of memnns (i do not buy their initial response to not testing against memory networks). 

I do have a few objections though. 

-- The details of the model are rather convoluted and the Section 2.1 is not very clearly written. In particular with the absence of the accompanying code the model will be super hard to replicate. I wish the authors do a better job in explaining the details as to how exactly the discrete operations are modeled, what is the role of the "row selector", the "scalar answer" and the "lookup answer" etc. 

-- The authors do a full attention over the entire database. Do they think this approach would scale when the data bases are huge (millions of rows)? Wish they experimented with larger datasets as well.

[Author Response · Arvind Neelakantan · 24 Dec 2016]
**Update from the authors**

We thank all the reviewers for the constructive feedback. We performed more experiments and added significant new material to the paper. To summarize:
1) We open-sourced the implementation of our model:

[Public Comment · Kelly Zhang · 10 Jan 2017]
**Questions regarding oracle and model design**

I have some questions about your paper:

1. Why is the oracle performance only about 50%? Is it limited by the type of operations you provide? Also is the oracle also only run for 4 timesteps? (If so would increasing the number of timesteps improve the oracle?)

2. In the paper that first introduces your neural programmer model ("Neural Programmer: Inducing Latent Programs with Gradient Descent"), you incorporate "logic" operations ("or" and "and"), why did you not include them as operations for this model?

[Final Decision · Program Chairs · 06 Feb 2017]
**ICLR committee final decision**

The paper applies a previously introduced method (from ICLR '16) to the challenging question answering dataset (wikitables). The results are strong and quite close to the performance obtained by a semantic parser. There reviewers generally agree that this is an interesting and promising direction / results. The application of the neural programmer to this dataset required model modifications which are reasonable though quite straightforward, so, in that respect, the work is incremental. Still, achieving strong results on this moderately sized dataset with an expressive 
 model is far from trivial. Though the approach, as has been discussed, does not directly generalize to QA with large knowledge bases (as well as other end-to-end differentiable methods for the QA task proposed so far), it is an important step forward and the task is already realistic and important.
 
 Pros
 
 + interesting direction
 + strong results on a interesting dataset
 
 Cons
 - incremental, the model is largely the same as in the previous paper